# Comparison between Egg Intake versus Choline Supplementation on Gut Microbiota and Plasma Carotenoids in Subjects with Metabolic Syndrome

**DOI:** 10.3390/nu14061179

**Published:** 2022-03-11

**Authors:** Minu S. Thomas, Marissa DiBella, Christopher N. Blesso, Olga Malysheva, Marie Caudill, Maria Sholola, Jessica L. Cooperstone, Maria Luz Fernandez

**Affiliations:** 1Department of Nutritional Sciences, University of Connecticut, Storrs, CT 06269, USA; minu.thomas@uconn.edu (M.S.T.); marissa.dibella@uconn.edu (M.D.); christopher.blesso@uconn.edu (C.N.B.); 2Department of Human Nutrition, Division of Nutritional Sciecnces, Cornell University, Ithaca, NY 14860, USA; ovm4@cornell.edu (O.M.); mac279@cornell.edu (M.C.); 3Department of Food Science and Technology, The Ohio State University, Columbus, OH 43210, USA; sholola.1@osu.edu (M.S.); cooperstone.1@osu.edu (J.L.C.); 4Department of Horticulture and Crop Science, The Ohio State University, Columbus, OH 43210, USA

**Keywords:** metabolic syndrome, gut microbiota, choline, TMAO, lutein, zeaxanthin

## Abstract

We previously demonstrated that intake of three eggs/d for 4 weeks increased plasma choline and decreased inflammation in subjects with metabolic syndrome (MetS). The purpose of the current study was to further explore the effects of phosphatidylcholine (PC) provided by eggs versus a choline bitartrate (CB) supplement on the gut microbiota, trimethylamine N-oxide (TMAO) formation, and plasma carotenoids lutein and zeaxanthin in MetS. This randomized, controlled crossover clinical trial included 23 subjects with MetS. Following a washout period of 2 weeks without consuming any choline-containing foods, subjects were randomly allocated to consume either three eggs/d or a CB supplement for 4 weeks (both diets had a choline equivalent of 400 mg/day). DNA was extracted from stool samples to sequence the 16S rRNA gene region for community analysis. Operational taxonomic units (OTUs) and the α-diversity of the community were determined using QIIME software. Plasma TMAO, methionine, betaine, and dimethylglycine (DMG) were quantified by stable isotope dilution liquid chromatography with tandem mass spectrometry. Plasma carotenoids, lutein, and zeaxanthin were measured using reversed-phase high-performance liquid chromatography. There were significant increases in plasma lutein and zeaxanthin after egg intake compared to the baseline or intake of CB supplement (*p* < 0.01). In contrast, TMAO was not different between treatments compared to the baseline (*p* > 0.05). Additionally, while diet intervention had no effects on microbiota diversity measures or relative taxa abundances, a correlation between bacterial biodiversity and HDL was observed. Following egg intake, the observed increases in plasma lutein and zeaxanthin may suggest additional protection against oxidative stress, a common condition in MetS.

## 1. Introduction

The link between egg-derived cholesterol and disease is subject to much investigation. Many population studies and clinical interventions show that egg intake does not increase the risk for cardiovascular diseases despite being rich in dietary cholesterol [1,2,3]. Recently, the Dietary Guidelines for Americans released by USDA in January 2016 removed the 300 mg/d limits for dietary cholesterol, shifting the focus to other components of eggs that may positively influence health [4]. Egg yolks contain the bioavailable forms of the potent antioxidant carotenoids lutein and zeaxanthin, which protect against the development of metabolic diseases by increasing the mRNA expression of antioxidant enzymes and decreasing pro-inflammatory cytokines [5]. Eggs positively modulated carotenoid concentrations and lipoprotein subclasses compared to egg substitutes in adult men following a carbohydrate-restricted diet [6]. HDL primarily transports lutein and zeaxanthin. This lipoprotein is increased by egg consumption, further raising the importance of eggs as a good food source for these carotenoids to protect against metabolic syndrome (MetS) [7]. However, mixed messages continue to advocate the health benefits of egg white after removing the yolk from the whole eggs. These controversial and untested pieces of information result in an ambivalent message to the public.

To further complicate the issue, eggs are an essential source of choline (as phosphatidylcholine), which is shown to produce TMAO (trimethylamine N-oxide) formed from the production of TMA (trimethylamine) by intestinal microbiota and followed to the conversion to TMAO by hepatic flavin-containing monooxygenases [8,9,10,11]. An elevated concentration of plasma TMAO is suggested to be a predictive biomarker and functional mediator in cardiovascular disease (CVD) in mice and humans [9]. This contributes to the uncertainty regarding the benefits of eggs, as TMAO is shown to promote atherogenesis through foam cell formation and due to its effects on increasing expression of scavenger receptors and a cluster of differentiation in macrophage receptors, CD36, and SR-A1, where the function of both interferes with reverse cholesterol transport [12,13,14]. Recent investigations suggest that TMAO and its derivatives contribute to platelet hyperreactivity and the resultant thrombosis, increasing the risk for cardiovascular diseases [15,16,17]. We previously demonstrated that when the consumption of three eggs/d is compared to the same amount of choline as a supplement (~400 mg), fasting plasma choline concentrations were higher with egg intake in a healthy young population [18]. Furthermore, we demonstrated that compared to the baseline, plasma TMAO was not increased by eggs or choline [18]. Hence, this study aimed to extend our knowledge by testing these dietary interventions on people at increased risk for chronic diseases, namely those classified with MetS.

Choline is an essential nutrient and can also be endogenously biosynthesized. Choline is oxidized to betaine, which serves as an intracellular osmolyte and methyl donor in the folate-independent remethylation of homocysteine to methionine, catalyzed by betaine-homocysteine methyltransferase [19], a reaction producing dimethylglycine. Higher intakes of choline and betaine are shown to be associated with lower plasma homocysteine levels [20,21,22]. Plasma homocysteine is an established risk factor for cardiovascular disease. These interactions of choline and the one-carbon metabolites intersects at multiple points between health and disease. Hence, this study aims to examine the effect of the source and its metabolites in MetS subjects.

Another emerging risk factor for consideration is the status of intestinal microbiota and its imbalance. Different gut microbiome profiles are found in MetS compared to healthy individuals, characterized by the proliferation of potentially harmful bacteria as well as the inhibition of beneficial bacteria [23]. We previously reported an increase in plasma choline in MetS subjects who consumed either 3 eggs/day or a a choline supplement; however, a decreased in inflammation was only observed after egg intake [24]. However, it is not clear how eggs contribute to TMAO formation in these individuals.

Therefore, the objective of this study was to determine whether the form in which choline is consumed (either as phosphatidylcholine from eggs or free choline from a supplement): (1) differentially affects plasma TMAO response, (2) modulates plasma lutein and zeaxanthin, and (3) alters the gut microbiota. We hypothesized that the choline provided as phosphatidylcholine (eggs) would lead to no changes in the TMAO response in this population. Additionally, we hypothesized that eggs containing highly bioavailable lutein and zeaxanthin would increase the concentrations of these carotenoids in the plasma, providing additional benefits to individuals with MetS.

## 2. Materials and Methods

### 2.1. Subjects and Experimental Design

The intervention scheme is presented in Figure 1. Twenty-three men and women classified with MetS according to the National Cholesterol Education Program (NCEP): Adult Treatment Panel (ATP) III criteria [25] were recruited to participate in this 13-week randomized, controlled cross-over clinical trial. Inclusion criteria for participation included 35–70 years, no liver disease, renal disease, diabetes, cancer, history of stroke or heart disease, or taking any glucose-lowering drugs or supplements. Other exclusion criteria, sample size, and the study design were described previously [24]. Subjects were given a list of foods high in choline and TMAO precursors to either avoid or be consistent with throughout the intervention as previously reported [24,26]. The subjects were free-living and were not supplied with any other food except for the eggs and choline supplement with no restrictions towards energy intake.

The study protocol was approved by the Institutional review board of the University of Connecticut. This clinical trial was registered to Clinicaltrials.gov (Protocol NCT03877003). Written informed consent was obrained from all subjects. All Baseline characteristics of the subjects were previously reported [24].

### 2.2. Blood Collection

After a 12 h fast, antecubital venous blood was collected in an EDTA-coated vacutainer, and the blood was immediately centrifuged at 2000× *g* for 20 min. The plasma was then aliquoted and was frozen at −80 °C before analyses.

### 2.3. Plasma Choline Metabolites

Plasma choline and its metabolites, betaine, dimethylglycine, and TMAO, were quantified by stable isotope dilution liquid chromatography with tandem mass spectrometry (LC/MS/MS), as previously described [27]. Plasma collected at the baseline and end of each intervention was used for these analyses.

### 2.4. Plasma Lutein and Zeaxanthin

The analysis of egg carotenoids in plasma was conducted using an Agilent 1260 HPLC-DAD and chromatographed using a C30 column (4.6 × 250 mm, 3 μm, YMC Inc., Wilmington, DE, USA) Five hundred μL of plasma was used to extract lutein and zeaxanthin [28,29,30]. Both carotenoids were quantified at 450 nm using authentic external standard curves.

### 2.5. Feces Collection

The University of Connecticut Microbial Center analyzed the fecal samples. Gut microbiota analysis was conducted via 16s V4 analysis. Fecal samples were collected from each subject at baseline and after completion of each dietary period using the microbial collection and stabilization OMNIgene GUT kits (DNA Genotek, Ottawa, ON, Canada). User instructions were provided to the subjects to effectively collect the sample at home a day before the first visit. Fecal samples were collected using sterile, disposable spatulas and placed in sterile tubes containing the stabilization liquid. Sixty-nine aliquots with duplicates were prepared and relocated to a deep freezer (−80 °C) until processing.

### 2.6. DNA Extraction, PCR Amplification, and Sequencing of Taxonomic Marker

DNA was extracted from 0.25 g of the fecal sample using the MOBIo PowerMag Soil 96-well kit (MoBio Laboratories, Inc., Carlsbad, CA, USA) according to the manufacturer’s protocol for the Eppendorf epMotion liquid handling robot. The DNA extracts were quantified using the Quant-iT PicoGreen kit (Invitrogen, ThermoFisher Scientific, Waltham, MA, USA). Partial bacterial 16S rRNA genes (V4, 0.8 picomoles each 515F and 806R with Illumina adapters and 8 basepair dual indices [16]) were amplified in triplicate 15 ul reactions using GoTaq (Promega, Madison, WI, USA) with the addition of 10 µg BSA (New England BioLabs, Ipswich, MA, USA). To overcome initial primer binding inhibition, as most primers do not match the template priming site, 0.1 femtomole 515F and 806R were added that do not have the barcodes and adapters. The PCR reaction was incubated at 95 °C for 2 min, the 30 cycles of 30 s at 95.0 °C, 60 s at 50.0 °C, and 60 s at 72.0 °C, followed by final extension as 72.0 °C for 10 min. PCR products were then pooled for quantification and visualization using QIAxcel DNA Fast Analysis (Qiagen, Hilden, Germany). PCR products were normalized based on the concentration of DNA from 350–420 bp then pooled using the QIAgility liquid handling robot. According to the manufacturer’s protocol, the pooled PCR products were cleaned using the Mag-Bind RxnPure Plus (Omega Bio-Tek, Norcross, GA, USA). Finally, the cleaned pool was sequenced on the MiSeq using a v2 2 × 250 basepair kit (Illumina, Inc., San Diego, CA, USA).

### 2.7. Stratification for Microbiota

Qiime 2 Bioinformatics (Scikit-Bio) was used to compare the significant bacterial phyla from each sample during the intervention, and correlations were determined with concentrations of plasma choline and TMAO.

### 2.8. Sequence Data Processing and Statistical Analyses

Sequences were demultiplexed using onboard bcl2fastq. Demultiplexed sequences were processed in Mothur v. 1.39.4 following the MiSeq SOP [31]. Emerged sequences that had any ambiguities or failed to meet length expectations were removed. Sequences were aligned to the Silva nr_v119 alignment [17]. Taxonomic identification of OTUs was derived using the RDP Bayesian classifier [32] against the Silva nr_v119 taxonomy database. Alpha and beta diversity statistics were calculated by taking the average of 1000 random subsampling to 10,000 reads per sample in Mothur. NMS and Permanova were run using the vegan package [33] in R 3.3.2. A subsampled species matrix was used for indicator species analysis [20]. Figures were drawn in R 3.3.2 using ggplot2 2.2.1 [34] and RColorBrewer 1.1–2.

### 2.9. Statistical Analysis

All variables were analyzed using SPSS for Windows Version 25 (IBM Corp). The level of significance for all tests was set at *p* < 0.05. The data are reported as mean ± SD. Choline metabolites, plasma TMAO, lutein, and zeaxanthin were analyzed by repeated-measures ANOVA where the repeated measure was the time (baseline and end of each dietary treatment) with Fisher’s LSD post hoc analysis. Finally, Pearson correlations were conducted between plasma biomarkers and microbiota diversity.

## 3. Results

### 3.1. Plasma Choline Metabolites

Significant increases in plasma choline from the baseline (7.9 ± 2.0 nmol/mL) to the end of intervention for both egg (9.9 ± 2.2 nmol/mL) and CB supplement periods (9.6 ± 2.1 nmol/mL) were previously reported, (*p* < 0.001) [24]. Similarly, there were significant increases in betaine and DGM from baseline to end of both interventions. At the same time, other metabolites such as methionine and TMAO showed no significant changes through the different periods, as shown in Table 1 and Figure 2.

### 3.2. Plasma Lutein and Zeaxanthin

Plasma concentrations of both lutein and zeaxanthin were significantly increased with the intake of three eggs/d compared to the baseline or to the intake of CB supplement (*p* < 0.01), as shown in Table 2. Although individuals were consuming three eggs/d, we observed no change in dietary lutein and zeaxanthin compared to baseline or CB supplement in agreement with previous reports [24].

### 3.3. Gut Microbiota

#### 3.3.1. Alpha-Diversity Indexes of the Gut Microbiota

The Shannon diversity index, which reports both abundance and evenness of the species present, was used to characterize species diversity among the treatment groups. We followed the usual approach to assess community changes by evaluating the change in alpha diversity (the variety and abundance of gut microbiota in a community) over time. This study establishes that these patterns of alpha diversity (within samples), Figure 3, did not differ markedly from comparisons between samples from the two choline sources among subjects (beta diversity), Figure 4.

#### 3.3.2. Beta-Diversity Indexes of the Gut Microbiota

Beta diversity was measured (similarity between multiple communities) to capture changes in community composition and diversity over time. The Bray–Curtis dissimilarity index was used to quantify differences in the overall taxonomic composition between the three groups (baseline, egg, CB). As shown in Figure 4, these comparisons did not reveal any statistically significant differences in the gut microbiota at baseline or after three eggs/d or CB supplementation. When focusing on the very abundant community members, time is still significant, explaining 5% of the variability. The interaction of time and treatment is borderline significant, explaining 2% of the variability.

#### 3.3.3. Correlation Analysis between Bacterial Diversity and Metabolic Parameters

Correlation analyses between bacterial diversity based on treatments and the subject’s clinical parameters were performed separately for each group of MetS patients (baseline, egg, CB) to identify associations between host clinical parameters and bacterial abundances. The results are summarized in Figure 5. We did not identify associations between bacterial diversity with biomarkers such as BMI, SBP, DBP, WC, plasma choline, glucose, TG, TC, and TMAO. Positive correlations were observed between bacterial diversity and plasma HDL concentrations (r = 0.79, *p* < 0.01)

#### 3.3.4. Comparison of Taxonomic Signatures in MetS Patients at Baseline and When Consuming Three Eggs/d Versus CB Supplementation

There were no significant changes between the baseline and the two treatment points at the taxonomic level, as shown in Figure 6a. *Ruminococcaceae* and *Lachnospiraceae,* belonging to Firmicutes phylum and *Bacteroides*, *Prevotella*, and *Alistipes* belonging to phylum Bacteroidetes, were the most abundant taxa observed in MetS patients throughout the 13 weeks while consuming three eggs/d or CB supplement. Verrucomicrobia, Proteobacteria, and Actinobacteria were the other prominent phyla observed in addition to the genus *Akkermansia*. While the relative abundance was similar with predominant families through the treatment period, the individual total microbial community percentage showed wide variation in the bacterial composition but had similar abundance in predominant families, as shown in Figure 6b. Overall microbiota diversity at the phylum level was relatively comparable across subjects during three eggs/d or CB supplementation with no significant changes in the ratios of Firmicutes to Bacteroidetes (F/B) but with a variable F/B ratio on individual levels (data not shown).

## 4. Discussion

The significant findings of this study are: (1) plasma TMAO was not increased after three eggs/d or CB supplementation for 4 weeks; (2) plasma lutein and zeaxanthin were increased after daily intake of eggs; (3) gut microbiota profiles of 23 MetS patients showed no significant difference in species richness within samples (alpha diversity) or across individuals at baseline or after treatment (beta diversity); (4) no correlations between plasma biomarkers and bacterial diversity were observed at baseline or after both treatments except for HDL.

One-carbon metabolites (choline, betaine, methionine, DGM) and TMAO are formed following dietary PC ingestion [13,35]. This study demonstrated that while choline [24], betaine, and DGM increased, there was no change in methionine and TMAO formation after intake of PC from eggs or CB supplementation compared to the baseline. The one-carbon metabolites formed are vital to metabolic pathways involved in essential physiological functions that do not include TMAO, such as the 1-C metabolism and lipid metabolism (e.g., fatty acid β-oxidation) [36,37]. The key finding that plasma TMAO did not change after dietary choline intake is at odds with some former studies indicating positive associations with TMAO concentrations and adverse cardiovascular outcomes [11,35,38,39,40,41,42], although past findings are inconsistent [43,44,45]. However, this result is consistent with the previously recognized mechanistic role of TMAO in accelerated atherosclerosis [46] and hints that increased production of the gut microbiota-dependent PC metabolite, TMAO, rather than the presence of or exposure to its substrates (e.g., choline, carnitine, betaine, etc.), is probably the overriding factor for the development of future cardiovascular events in humans. It is also possible that TMAO concentrations reflect rather than cause increased CVD risk [47]. Thus, this study highlights that similar to healthy young subjects consuming ∼400 mg/d choline either via eggs or choline supplementation, [18], plasma TMAO concentration is not affected in MetS subjects, thereby confirming that consumption of three eggs/d for 4 weeks does not increase the risk for CVD via TMAO responses due to dietary choline intake.

While insulin resistance plays a pivotal role in initiating, propagating, and modulating the pathologic manifestations of MetS, oxidative stress is also identified as another major contributor [48]. Oxidative stress plays a critical role in the pathophysiology of type II diabetes and cardiovascular disease [49,50,51]; hence antioxidant status among individuals with MetS who are at high risk for developing these conditions is critical to be evaluated [52]. MetS are associated with suboptimal levels of several antioxidants, including lutein and zeaxanthin [53]. While the association of carotenoids with MetS is unclear, higher serum concentrations of these carotenoids are associated with a lower prevalence of MetS and fewer abnormal MetS criteria in Chinese adults [54]. An Australian cross-sectional study shows a similar pattern where carotenoids decreased significantly as the number of components of the MetS increased [55]. These carotenoids are potent scavengers of free radicals in vitro. It is proposed using microglial cells that, due to their polarity and extended conjugated double bonds, lutein and zeaxanthin attenuate the development of neurodegenerative diseases by promoting the mRNA expression of antioxidant enzymes lowering pro-inflammatory cytokines [56]. These anti-inflammatory and beneficial antioxidant properties of lutein and zeaxanthin may have similar beneficial effects on inflammatory biomarkers and metabolic risk factors of MetS.

It is debated whether the low concentration of plasma lutein and zeaxanthin in MetS results from lower consumption of dietary carotenoids or because of its increased utilization due to oxidative stress effects [57,58]. Eggs are already established as a highly bioavailable source of lutein and zeaxanthin [59], as they contain lipids that enable carotenoid absorption [60]. Oxygenated carotenoids are primarily transported by HDL particles which are increased after egg intake [7]. We previously reported that there were no statistical differences in the mean dietary intake of lutein and zeaxanthin in our MetS population after consuming three eggs/d (226 ± 1835 mg/d) and after consuming an equivalent amount of choline from CB supplement (2600 ± 4222 mg/d) for 4 weeks [24]. The significant increases in plasma lutein and zeaxanthin after egg intake, compared to the baseline or to the intake of a supplement (*p* < 0.01), is consistent with previous findings [7,61,62] that eggs provided bioavailable carotenoids even though the average dietary intake of carotenoids were similar during both periods.

Current evidence suggests that MetS is accompanied by an imbalance of the normal gut microbiota (dysbiosis), where the metabolites produced disrupts the gut mucosal barrier leading to gut permeability and metabolic endotoxemia, possibly initiating chronic low-grade inflammatory response, which could trigger the development of insulin resistance, promoting the progression of MetS [23,63,64,65]. The pathophysiology of MetS is complex; its progression is determined and contributed by multiple factors leading to metabolic dysfunction and thus cannot be limited, as explained above, and the mechanisms are still unclear [23,66,67,68]. There is little evidence on whether the beneficial effects of eggs on MetS can be linked to their potential impact on the gut microbiota and intestinal integrity. Thus, our pilot study aimed to provide a clearer understanding of the topic. Our result revealed no remarkable changes in the gut microbiota at baseline and after consuming three eggs/d or CB supplementation for 4 weeks each. Except for the correlation between bacterial diversity and HDL cholesterol, the other parameters directly related to MetS (such as BMI, SBP, DBP, WC, triglyceride, glucose, and insulin) were not significantly different between the treatments. This correlation indicates the association of gut microbiota with lipid metabolism [69].

Several findings indicated that the gut microbiota can modify blood lipid composition, particularly cholesterol, through their role in bile acid metabolism and the generation of microbial products [70,71,72,73]. In a subset of the LifeLines population-based cohort, Fu et al. proved significant associations between proportions of specific gut microbiota with HDL cholesterol and TG levels in subjects independent of BMI, suggesting a role of gut microbes in altering host lipid metabolism [74]. The positive correlation of HDL cholesterol in our study is consistent with these previous findings. The high variability in an individual’s gut microbiome and the resultant metabolic changes make it harder to form associations between metabolic syndrome parameters and gut microbiome diversity and abundance. The cohort of patients proved this diversity across individuals among gut microbiota profiles in MetS patients at a high risk of cardiovascular events (CORE)-Thailand earlier [75]. These MetS patients were classified based on their similarity into three groups or enterotypes and showed several associations between species abundance and metabolic parameters that are enterotype specific. Similar to our findings, there were no strong associations between the abundance of any genus and clinical parameters of MetS after correlation analyses [75].

Our findings did not support the hypothesis that the gut microbiota population will be different based on the form of dietary choline and the metabolites produced. Notably, the alpha diversity (Shannon diversity and observed OTUs) and the beta diversity (Bray–Curtis dissimilarity) of individual diet were not significantly correlated with gut microbiota, despite subtle changes in the community composition. This suggests that either the changes in the composition of the gut microbiota was transient, that the gut microbiome adjusted to the changes before 4 weeks, or that the intervention had a lasting effect on the gut microbiota beyond the intervention stage, proving the washout period of 3 weeks as ineffective while carrying forward the effect or simply the 4 weeks was not enough to bring about a significant change. With the lack of similar studies to compare, it is difficult to come to a single conclusion. Our study is the first to highlight this relationship between choline in diet and gut microbiota composition in MetS subjects. It suggests that the species richness within individual gut microbiota did not modify based on choline source. Individuals with MetS harbor more similar gut microbiota irrespective of treatment time. The diet of the subjects, although ad libitum with no calorie restrictions, except for a choline-free diet, thus did not drastically change the gut microbiota through the treatment period.

Obesity and MetS are associated with a higher intestinal F/B ratio than lean or healthy obese people [63,76,77]. In lean subjects, the consumption of a calorie-restricted diet is accompanied by a reduction of body weight and a shift from the high F/B ratio to a lower value [63]. In contrast, energy-rich diets were reported to increase the proportion of intestinal Firmicutes in both humans and mice, suggesting that dietary ingredients or endogenous metabolites (e.g., bile acids) secreted into the gut lumen in response to these diets were responsible for this occurrence. However, several discrepancies are not understood as not many human trials reported similar proportions [67,78]. While our study showed the Firmicutes and Bacteroidetes represented the dominant portion of the gut microbiota phyla, which usually forms 90% of the gut microbiota [79], no significant findings corresponded to the F/B ratio. *Alistipes*, although present at a relatively low rate compared to other genus members of the Bacteroidetes phylum (*Bacteroides, Prevotella*), are highly relevant in dysbiosis and metabolic diseases were shown to have a significant role in inflammation and related pathophysiology as per emerging reports [80]. These findings provide the guideline for the gut microbiota composition in a MetS population even though no significant changes were derived after consuming the two choline sources in their diet. *Akkermansia muciniphilia* is among the most abundant bacterial species in the human gut and is potentially a beneficial microbe in treating metabolic diseases, especially MetS [81,82,83]. A cross-sectional study in MetS patients has proven that the relative abundance of *Akkermansia* was inversely associated with the risk of MetS. A 1% increase in *Akkermansia* reduced the risk of MetS by ~0.3% [84]. Microbial interaction analyses showed that *Ruminococcaceae* and *Lachnospiraceae* are the predominant bacterial families correlated with *Akkermansia* abundance and influenced by the *Akkermansia*-MetS association [84]. Hence, these in our study portray the gut microbiome’s attempt to balance the dysbiosis was caused due to pathogenic factors driving MetS.

Significant interindividual variations in the gut microbiome contribute to the overall interpretation of metabolites. They are traditionally related to genetic factors, and the remaining portion of their variance is accredited to environmental factors. A manifold of factors mediates the interplay of gut microbiota with the host metabolism; hence, extended studies are required to further our knowledge about gut microbiota in MetS patients. There are some limitations in this study, the sample size was calculated based on plasma choline levels, but it is possible that if we had more subjects, we might have found significant data in the microbiota.

## 5. Conclusions

Choline in this study was provided by two distinct sources: phosphatidylcholine from eggs or choline bitartrate, which resulted in significant increases in plasma choline [24] without changes in TMAO. However, the microbiota diversity characteristic of metabolic syndrome patients was not altered by including choline in the diet or the carotenoids lutein and zeaxanthin present in eggs. Although plasma values of choline in both dietary treatments and lutein and zeaxanthin after the egg treatment were increased, indicating compliance, the microbiota was not modified, meaning that these individuals need more extended time for this type of intervention to obtain meaningful changes in the microbiota. Further studies with extended periods or different populations need to be conducted to understand the role of dietary choline, lutein, and zeaxanthin on microbiota diversity in subjects with metabolic syndrome.

## Figures and Tables

**Figure 1 nutrients-14-01179-f001:**
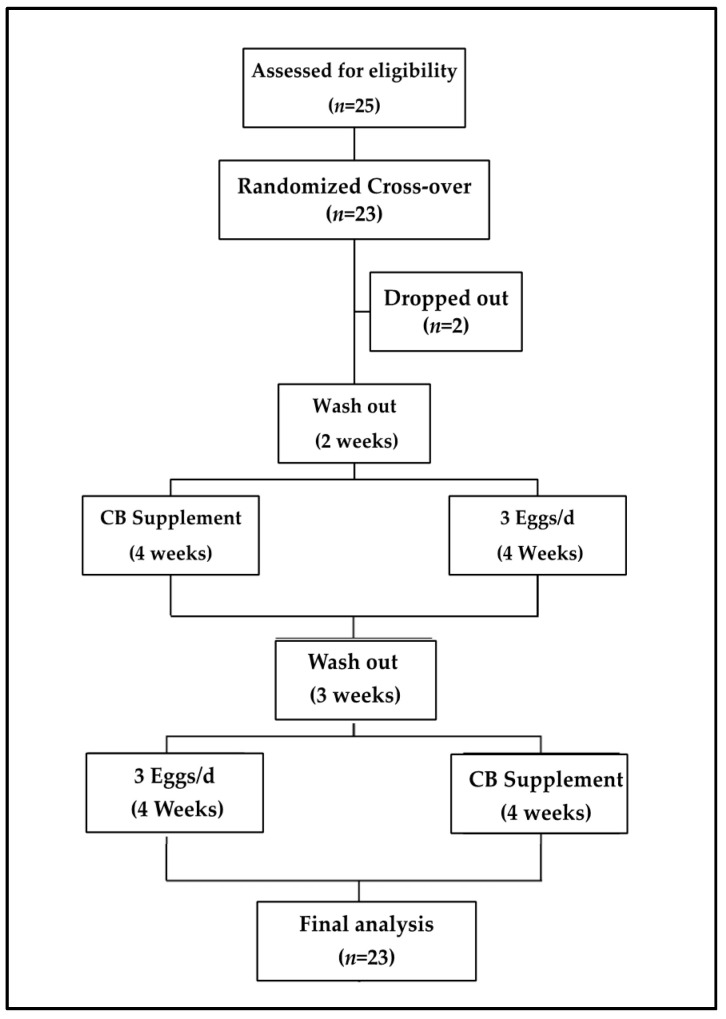
Experimental design. Subjects underwent a 2 week washout period before being randomly allocated to either 3 eggs/d or CB supplement (~400 mg) for 4 weeks. Following a 3 week washout period, they were assigned to the alternate treatment. Blood to measure choline, TMAO, and other metabolites and feces was collected at baseline and the end of each dietary period.

**Figure 2 nutrients-14-01179-f002:**
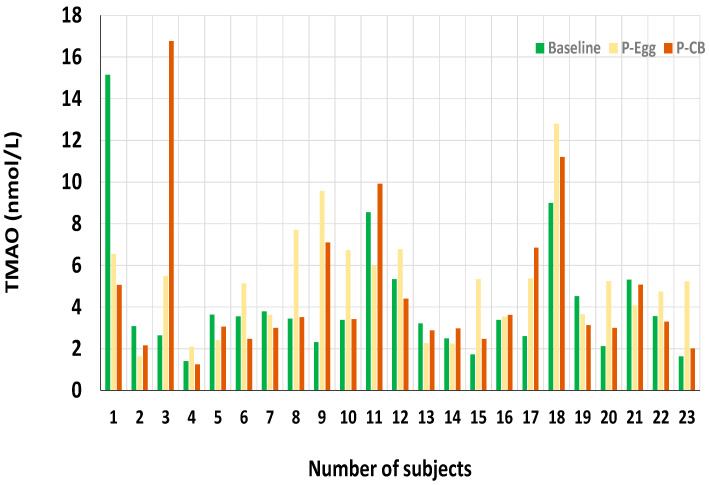
Plasma concentration of TMAO at baseline and post-consumption of 3 eggs/d (P-Egg) or CB supplementation (P-CB) for 4 weeks. Data is shown for the 23 subjects at each time point: baseline, after the egg intervention (P-Egg), and after the choline supplement (P-CB).

**Figure 3 nutrients-14-01179-f003:**
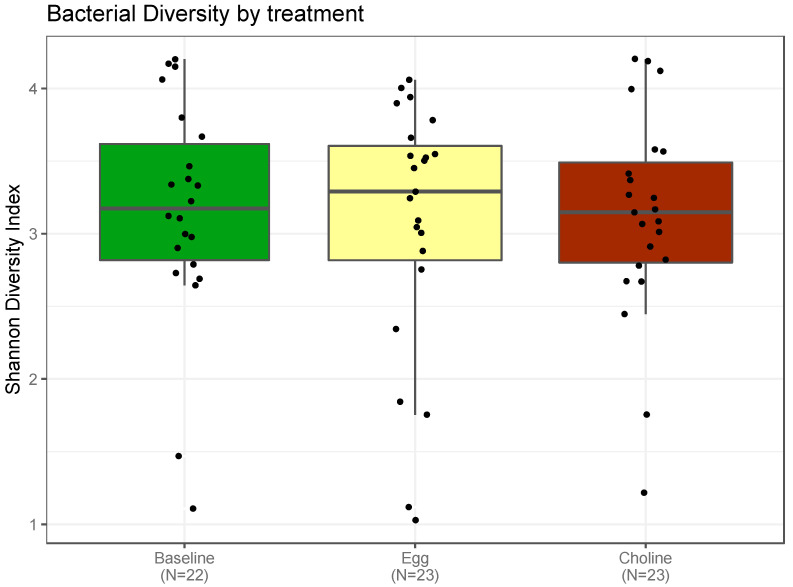
Alpha diversity was measured by Shannon diversity index in subjects with MetS at baseline and post-consumption of 3 eggs/d (P-Egg) or CB supplementation(P-CB) for 4 weeks. The solid bar indicates the mean, and the scattered point represents the variation among subjects. No significant changes were observed between the treatments. One of the samples for baseline due to errors during the preparation could not be included in this figure.

**Figure 4 nutrients-14-01179-f004:**
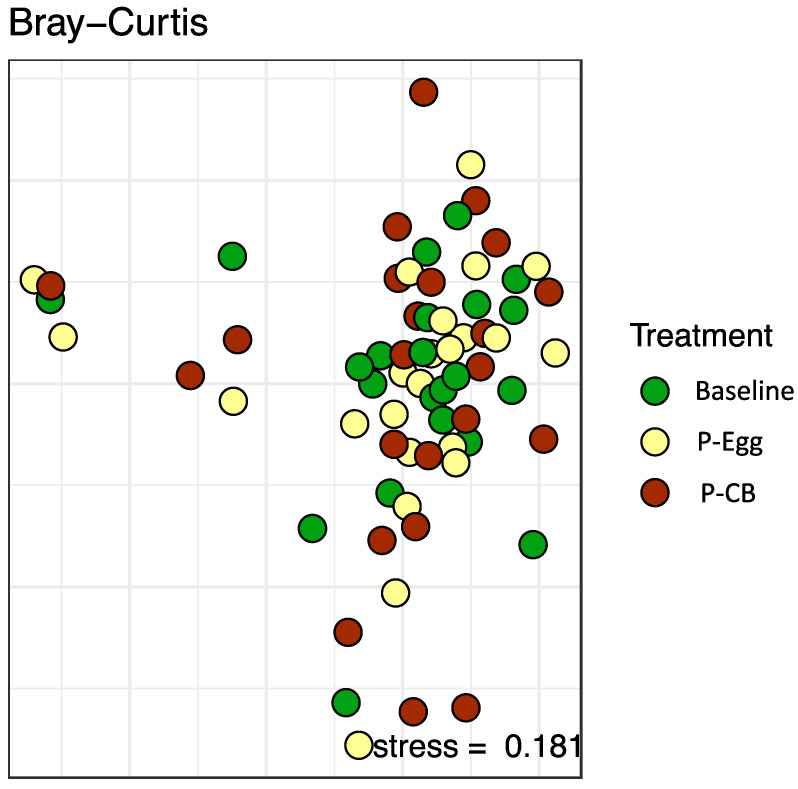
Nonmetric multidimensional scaling of the Bray–Curtis dissimilarity index, showing the relationship between individual samples based on the species and their abundances at baseline, post 3 eggs/d, and post-CB supplementation for 4 weeks.

**Figure 5 nutrients-14-01179-f005:**
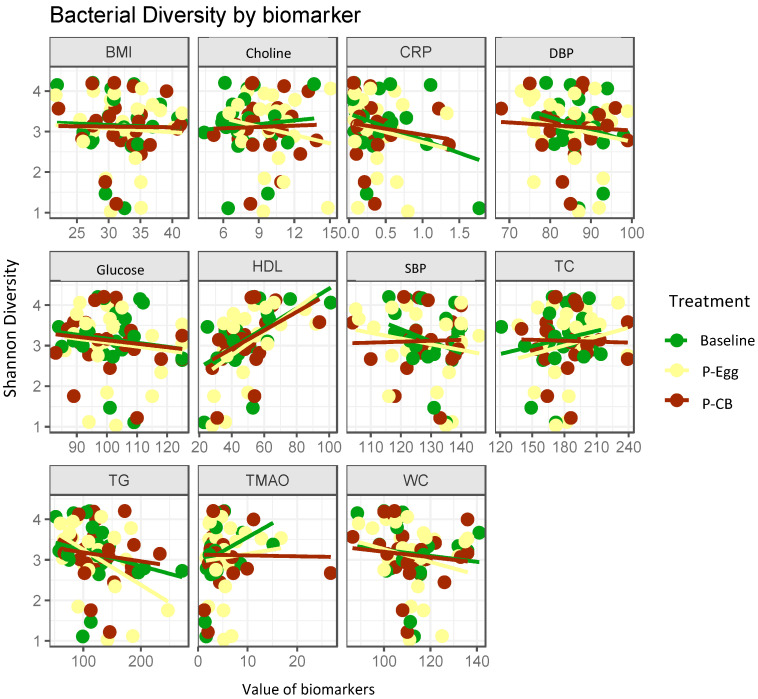
Correlations between plasma biomarkers and bacterial diversity based on treatments. BMI = body mass index, CRP = C reactive protein; DBP = diastolic blood pressure; HDL = high density lipoprotein; SBP = systolic blodd pressure; TG = triglycerides, TC = total cholesterol; TMAO = trimethylamince oxid; WC = waist circumference at baseline and after consumption of 3 eggs/d (Egg) or choline bitartrate (Choline) supplementation for 4 weeks. Only the correlation between bacteria diversity and plasma HDL cholesterol was found to be significant (r = 0.79, *p* < 0.01).

**Figure 6 nutrients-14-01179-f006:**
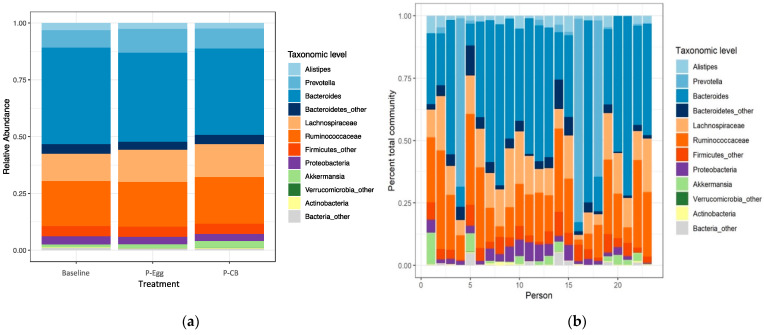
Relative abundance of the gut microbiota (**a**) based on treatments (**b**) and individual average throughout the intervention.

**Table 1 nutrients-14-01179-t001:** Plasma concentration of choline metabolites at baseline and after consumption of 3 eggs/d (P-Egg) or after choline bitartrate (P-CB) supplementation for 4 weeks ^1^.

Parameter	Baseline	P-Egg	P-CB
TMAO (nmol/mL)	4.16 ± 3.05	5.14 ± 2.5	5.15 ± 5.3
Methionine (nmol/mL)	29.4 ± 4.0	29.8 ± 4.0	29.9 ± 5.2
Betaine (nmol/mL) ^2^	37.6 ± 14.8 ^a^	43.0 ± 14.7 ^b^	43.5 ± 18.7 ^b^
DGM (nmol/mL)	2.4 ± 0.6 ^a^	2.8 ± 0.7 ^b^	2.8 ± 0.6 ^b^

^1^ Data are presented as mean ± SD for *n* = 23 subjects. ^2^ Numbers in the same row (a, b) with different superscripts are significantly different (*p* < 0.01).

**Table 2 nutrients-14-01179-t002:** Plasma lutein and zeaxanthin in subjects at baseline and post (P) intervention after either egg (P-Egg) or choline bitartrate (P-CB) supplement for 4 weeks ^1^.

Parameter	Baseline	P-Egg	P-CB
Plasma lutein (nmol/L) ^2^	495.1 ± 235 ^a^	681.6 ± 351 ^b^	527.7 ± 283 ^a^
Plasma zeaxanthin (nmol/L)	64.8 ± 29 ^a^	113.8 ± 46 ^b^	67.3 ± 27 ^a^

^1^ Data are presented as mean ± SD for *n* = 23 subjects. ^2^ Numbers in the same row with different superscripts (a, b) are significantly different (*p* < 0.01).

## Data Availability

Data is available upon request to the Principal Investigator.

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
