# Peer review of "Comparison between Egg Intake versus Choline Supplementation on Gut Microbiota and Plasma Carotenoids in Subjects with Metabolic Syndrome"

_nutrients, 2022, doi:10.3390/nu14061179_

Round 1
Reviewer 1 Report
In the present manuscript, the authors conducted an association study of egg and metabolites pattern. The authors explore the effects of PC provided by eggs vs. CB supplement on gut microbiota, TMAO formation, and plasma carotenoids lutein and zeaxanthin in MetS. The results showed that plasma lutein and zeaxanthin significantly increased after egg intake, compare to baseline or CB supplement intake. Plasma TMAO was not increased in both groups. No correlations between plasma biomarkers and bacterial diversity were observed at baseline or after both treatments except for HDL.
This manuscript was well-thought and well-executed, and it could have a significant impact of the field. However, there are both major and minor concerns that should be addressed before accepting for publication in Nutrients.
Major:
- The abstract should be revised to be concise and attractive.
- Sample numbers of each group was not enough, this may lead to nonsignificant analysis and mislead the readers. Otherwise, the rational of adopting this sample number should be included.
- Initial characteristics of subjects was not shown.
- Table 1 and 2 should be presented as dot plots like Figure 5 since these were the main results of this manuscript.
- Figure 3. Please include explanation of n=22 for baseline, while n=23 for P-Egg and P-CB.
- Figure 5. It seems the labels of each plot were artificially added, such as “Choline, DBP, Glucose, SBP”, since the font size of these labels were not the same as others. Please show the original data.
Minor:
- Figure 2. Please show all data points on top of bar plot.
- Line 212: “(CB)” should be “P-CB”.
- Figure 3. Please keep the color code consistent as other figures (Green for Baseline and Yellow for P-Egg).
- Line 261: “r – 0.79” should be “R = 0.79”.
- Please show full name of HDL.
- Line 315: “3eggs /d” should be “3 eggs/d”.
- Line 329: “in vitro” should be “in vitro”.
- Line 378: “Mets patients” should be “MetS patients”.
- The font size of references was not consistent.
Author Response
We thank the reviewer for their thorough review of our manuscript to make it more clear and for catching all the places where editing was needed.
- The abstract should be revised to be concise and attractive R. We have modified the abstract to make it more attractive
2. Sample numbers of each group was not enough, this may lead to nonsignificant analysis and mislead the readers. Otherwise, the rational of adopting this sample number should be included.
- R. We did a sample size calculation based on plasma choline. We needed 20 subjects to see differences in this parameter based on previous studies. This was previously reported (reference #24).3. Initial characteristics of subjects was not shown.
- R. The characteristics of the subjects had already been previously reported (reference 24). We cannot duplicate the data. Here is the Table with the baseline characteristics from reference 24.
Table 2. Initial Characteristics of Subjects
|
Parameter |
Values |
|
Age (years) |
55.2 ± 8.9 |
|
Gender Females (%) |
65 |
|
Waist Circumference (cm) |
113.3 ± 12.4 |
|
Systolic Blood Pressure (mm Hg) |
129.3 ± 7.3 |
|
Diastolic Blood Pressure (mm Hg) |
87.5 ± 4.8 |
|
HDL cholesterol (mg/dL) |
47.7 ± 17.4 |
|
Triglycerides (mg/dL |
129.5 ± 57.6 |
|
Glucose (mg/dL) |
102.6 ± 10.5 |
Values are expressed as mean ± Standard deviation for n = 23 subjects
4. Table 1 and 2 should be presented as dot plots like Figure 5 since these were the main results of this manuscript.
- R. Thank you’re your comment. However, we prefer to show the data in a numerical form in a Table. We do not want to increase the number of figures in this manuscript
5. Figure 3. Please include explanation of n=22 for baseline, while n=23 for P-Egg and P-CB.
- R. We have included an explanation, which was due an error in the preparation of the sample
6. Figure 5. It seems the labels of each plot were artificially added, such as “Choline, DBP, Glucose, SBP”, since the font size of these labels were not the same as others. Please show the original data.
- R. The reviewer is right, the labels were added to make them more clear. However, these figures do represent the original data.
Minor:
- Figure 2. Please show all data points on top of bar plot.
R. Figure 2 has been redone and it now showing plasma TMAO levels in every single point. We believe this figure is more clear
2. Line 212: “(CB)” should be “P-CB”.
- R. The figure legend says: or CB supplementation (P-CB). This is correct as it is.
3. Figure 3. Please keep the color code consistent as other figures (Green for Baseline and Yellow for P-Egg).R. We appreciate this comment. I am glad that the reviewer noticed this mistake that was overlooked by 8 authors. We have corrected the figure
4. Line 261: “r – 0.79” should be “R = 0.79”.R. Yes, it has been corrected
5. Please show full name of HDL.. R. Full name is now shown
6. Line 315: “3eggs /d” should be “3 eggs/d”. R. Modified as requested
7. Line 329: “in vitro” should be “in vitro”. R. It has been corrected
8. Line 378: “Mets patients” should be “MetS patients”. R. It has been corrected
9. The font size of references was not consistent. R. The problem has been corrected
Reviewer 2 Report
Dear Authors:
The authors have carried out a randomized, controlled cross-over clinical trial tittled “Comparison Between Egg Intake Versus Choline Supplementation on Gut Microbiota and Plasma Carotenoids in Subjects with Metabolic Syndrome”. The aim of this study was to a determine whether the form in which choline was consumed could affect the plasma trimethylamine N-oxide response, the modulation of plasma lutein and zeaxanthin and could alter gut microbiota.
The study is well designed, comprehensive, rigorous, and scientifically correct,- and incorporates novel aspects in the relationship between choline in diet and gut microbiota composition in metabolic síndrome patients. From an academic point of view the manuscript is original and gives good knowledge of other modulating factors involved in the development of oxidative stress and metabolic síndrome.
The figures are very illustrative and well designed.
Nevertheles, some considerations need to be taken into account:
- Given the frequency of metabolic syndrome in the clinic, the small sample used in the study is striking (n: 23 subjects). This study would surely have benefited from a larger sample size, which would undoubtedly strengthen its results and conclusions.
- Although it is partially reflected in the discusión, what is the hypothesis that in your opinion is more consistent regarding the non-modification of the microbiota after the study leaving aside the time variable?
- Line 388, Replace “The” with “This”
- I missed in the discussion the description of possible limitations of the study
- Bibliographic reference number 46 seems to be misreferenced. The Journal should must be referenced in the text (Curr Dev Nutr. 2020 Dec 11;5(1):nzaa179.doi: 10.1093/cdn/nzaa179. eCollection 2021 Jan.)
Kind regards
Author Response
The authors have carried out a randomized, controlled cross-over clinical trial tittled “Comparison Between Egg Intake Versus Choline Supplementation on Gut Microbiota and Plasma Carotenoids in Subjects with Metabolic Syndrome”. The aim of this study was to a determine whether the form in which choline was consumed could affect the plasma trimethylamine N-oxide response, the modulation of plasma lutein and zeaxanthin and could alter gut microbiota.
The study is well designed, comprehensive, rigorous, and scientifically correct,- and incorporates novel aspects in the relationship between choline in diet and gut microbiota composition in metabolic síndrome patients. From an academic point of view the manuscript is original and gives good knowledge of other modulating factors involved in the development of oxidative stress and metabolic síndrome.
The figures are very illustrative and well designed.
Thank you for your kind comments
Nevertheles, some considerations need to be taken into account:
- Given the frequency of metabolic syndrome in the clinic, the small sample used in the study is striking (n: 23 subjects). This study would surely have benefited from a larger sample size, which would undoubtedly strengthen its results and conclusions.
- We agree with the reviewer in this point. However, our main outcome was plasma choline and we determined that we needed only 20 subjects to see differences in plasma choline, which was our main outcome (reference 24).
- Although it is partially reflected in the discusión, what is the hypothesis that in your opinion is more consistent regarding the non-modification of the microbiota after the study leaving aside the time variable?.
- R. We hypothesized that both plasma choline and lutein and zeaxanthin would be changed by egg intake. We did not hypothesize that we would see changes with the short duration of the study. This is why we put that as an objective rather than a hypothesis. In the discussion we explained why we think we did not see changes
- Line 388, Replace “The” with “This”
- It has been done
- I missed in the discussion the description of possible limitations of the study
- Possible limitations have been added
- Bibliographic reference number 46 seems to be misreferenced. The Journal should must be referenced in the text (Curr Dev Nutr. 2020 Dec 11;5(1):nzaa179.doi: 10.1093/cdn/nzaa179. eCollection 2021 Jan.)
- Thank you for catching up this problem. It has has been corrected
Round 2
Reviewer 1 Report
The authors have revised several places; however, the manuscript still reads non-rigorous, needs to be revised. For example:
- Replotted figure 2 could not show the significance of the results, it could be better to plot like figure 3.
- Figure 4 did not show axis labels.
- The statistical p-value sometime showed as “P” (Tables 1&2, line 216, etc.), sometime showed as “p” (lines 194, 203, 260, etc.), please read through carefully and revised to be consistent.
- Ref [70] should be “Kriaa, A.; Bourgin, M.; Potiron, A.; Mkaouar, H.; Jablaoui, A.; Gérard, P.; Maguin, E. and Rhimi, M. Microbial impact on cholesterol and bile acid metabolism: current status and future prospects. Journal of lipid research, 2019 60, doi:10.1194/jlr.R088989.”
Author Response
- Replotted figure 2 could not show the significance of the results, it could be better to plot like figure 3.
- R. We prefer to keep Figure 2 as the new figure because it shows the TMAO levels for each individual in each point. The statistics are already presented in Table 1 that shows that TMAO levels were not different among groups, The purpose of this figure is to show the intra-individual variation of each of the 23 subjects so we believe this is more informative.
- Figure 4 did not show axis labels.
The figure is non-metric so it is appropriate to display with no labels. Other authors have published their figures this way (Wilkes et al. Microbial Technology 2018; 1-4 ; Yeoman et al. Scientific Reports 2018: 8:3197
- The statistical p-value sometime showed as “P” (Tables 1&2, line 216, etc.), sometime showed
as “p” (lines 194, 203, 260, etc.), please read through carefully and revised to be consistent.
- all are now shown as “p” to be consistent
- Ref [70] should be “Kriaa, A.; Bourgin, M.; Potiron, A.; Mkaouar, H.; Jablaoui, A.; Gérard, P.; Maguin, E. and Rhimi, M. Microbial impact on cholesterol and bile acid metabolism: current status and future prospects. Journal of lipid research, 2019 60, doi:10.1194/jlr.R088989.”
Ref 70 is now OK